# The Notch1 intracellular domain orchestrates mechanotransduction of fluid shear stress

Tania Singh[1,2], Kyle A Jacobs[1,4], William J Polacheck[5,6], Matthew L Kutys[1,2,3,4]

**Hemodynamic shear stress regulates endothelial phenotype through activation of Notch1 signaling, yet the mechanistic basis for this activation is unclear. Here, we establish a fluid shear stress–dependent mechanism of Notch1 activation that is distinct from canonical ligand trans-endocytosis. Application of unidirectional laminar flow triggers the rapid spatial polarization of full-length Notch1 heterodimers into downstream membrane microdomains. Unlike canonical transactivation, Notch1 receptors are cis-endocytosed into the receptor-bearing cell within polarized microdomains. We discover that the Notch1 intracellular domain critically orchestrates receptor polarization and proteolytic cleavage in response to flow, but is dispensable for canonical ligand transactivation. Shear stress increases intracellular domain interaction with annexin A2 and caveolar proteins, which control Notch1 cis-endocytosis and proteolytic activation. These findings define a flow-specific Notch1 mechanotransduction mechanism linking receptor polarization and endocytosis with proteolytic activation and illuminate a new pathway by which mechanical forces integrate with Notch receptor activation.**

## Introduction

Blood flow is a key determinant of vascular development and postnatal vascular homeostasis (Hahn & Schwartz, 2009; Campinho et al, 2020). Hemodynamic shear stress, the frictional force exerted by blood flow on endothelial cell lining vessels, is transduced into biochemical signaling to regulate endothelial phenotype. Disruption of this mechanotransduction drives endothelial dysfunction and contributes to myriad cardiovascular diseases including atherosclerosis, stroke, and vascular malformations (Davies, 2009; Deng et al, 2025). A central endothelial pathway stimulated by shear stress is Notch1 receptor signaling. Shear stress–induced Notch1 signaling in arterial and microvascular endothelia regulates endothelial fate specification, cell cycle progression, calcium signaling, and vessel morphogenesis and barrier function (Krebs et al, 2000; Masumura et al, 2009; Jahnsen et al, 2015; Fang et al, 2017; Mack et al, 2017; Polacheck et al, 2017). Despite the deep appreciation of the importance of endothelial behaviors controlled by shear stress and Notch1 signaling, a mechanistic understanding of how the Notch1 receptor is mechanically activated by shear stress is critically lacking. This knowledge gap is significant, as it offers the potential for new therapeutic targets for angiogenesis, barrier function regulation, and other Notch-associated vasculopathies.

Notch transmembrane receptors are canonically activated through a sender–receiver mechanism involving trans interaction with transmembrane ligands expressed by adjacent cells. In addition to receptor–ligand engagement, mechanical force on Notch is required for activation (Gordon et al, 2015; Bray, 2016; Siebel & Lendahl, 2017). This force is required to induce conformational changes that sensitize the receptor to sequential proteolytic cleavages and can arise via trans-endocytosis of the ligand–receptor pair into the ligand-expressing cell (Kopan & Ilagan, 2010; Gordon et al, 2015). The Notch extracellular S2 site is first cleaved by a disintegrin and metalloproteinase (ADAM) to release the extracellular domain (ECD) (Kovall et al, 2017). Subsequently, the γ-secretase complex cleaves Notch at the intramembrane S3 site resulting in release of the intracellular domain (ICD), which functions as a pleotropic transcriptional coactivator (Kovall et al, 2017). Endothelial Notch1 activation by shear stress has been reported to require Piezo1 ion channel activity and ADAM10 (Caolo et al, 2020). However, whether shear stress drives endothelial Notch1 receptor proteolysis and signaling activation through mechanically gated, ligand transactivation or an alternative mechanism is unknown.

Notch1 is expressed broadly, and accumulating evidence indicates that Notch receptor activation and downstream function are linked to cell adhesions and cell cortex mechanics in distinct cellular contexts (Lowell & Watt, 2001; Crowner et al, 2003; Khait et al, 2016; Shaya et al, 2017; Hunter et al, 2019; Priya et al, 2020; Falo-Sanjuan & Bray, 2021; Kwak et al, 2022; White et al, 2023). One emerging theme is that subcellular spatial regulation of Notch is a

[1]Department of Cell and Tissue Biology, University of California San Francisco, San Francisco, CA, USA  [2]Joint Graduate Program in Bioengineering, University of California, Berkeley and San Francisco, San Francisco, CA, USA  [3]Cardiovascular Research Institute, University of California San Francisco, San Francisco, CA, USA  [4]Biomedical Sciences Graduate Program, University of California San Francisco, San Francisco, CA, USA  [5]Lampe Joint Department of Biomedical Engineering, University of North Carolina at Chapel Hill, Chapel Hill, NC, USA  [6]North Carolina State University, Raleigh, NC, USA

Correspondence: matthew.kutys@ucsf.edu

critical mechanism controlling receptor proteolysis and activation. Specifically, localizing Notch receptors to membrane microdomains enriched for γ-secretase is a key step before receptor S3 cleavage and ICD release (Kwak et al, 2022). In epithelia and neurons, adherens junctions organize size-selective, membrane proteolytic hotspots containing flotillin and γ-secretase that limit Notch signaling by size-excluding the bulky Notch ECD (Kwak et al, 2022). In *Drosophila*, Notch ubiquitination and partitioning into distinct endosomal membrane microdomains enriched for cholesterol or clathrin/ESCRT-0 result in ligand-independent Notch activation (Shimizu et al, 2024) and Notch1 activity is limited by the formation of lateral membrane contacts and adherens junctions during cellularization of the embryonic syncytium (Falo-Sanjuan & Bray, 2021).

Postnatal endothelial cells express Notch1 and Delta-like (Dll) and Jagged (Jag) ligands, so we sought to identify molecular control systems governing Notch1 activation by hemodynamic shear stress. We identify that Notch1 activation by unidirectional laminar flow is the product of a specific endothelial mechanotransductive response involving spatial reorganization of the receptor into polarized plasma membrane microdomains located downstream of flow. Full-length Notch1 heterodimers translocate to flow-polarized domains where they are rapidly cis-endocytosed into the receptor-bearing cell. Interestingly, flow-dependent Notch1 polarization and cis-endocytosis are organized by interactions with the Notch1 ICD. We determine the ICD governs Notch1 proteolysis and activation in response to flow or on immobilized DLL4 substrate, but is dispensable for Notch1 activation by DLL4 in a canonical sender–receiver trans-activation model. Using unbiased proteomics, we identify that flow promotes the association of ICD with annexin A2 and caveolar proteins, which are essential for Notch1 activation by shear stress through the coordination of receptor endocytosis and proteolysis, respectively. Altogether, our findings establish new spatial and molecular mechanisms that govern the activation of endothelial Notch1 by flow, and more broadly provide new avenues by which Notch receptors may be engaged by cell-generated forces.

# Results

## DLL4 is required, but not sufficient, for Notch1 activation by shear stress

Application of unidirectional laminar shear stress (~20 dyne/cm$^2$) to primary human dermal blood microvascular endothelial cells (hdBECs) results in elevated γ-secretase–mediated cleavage of the Notch1 ICD (as measured by a cleavage-specific Notch1 V1754 antibody) at 1 h and is sustained for 24 h under flow (Fig S1A). Hereafter, we refer to this receptor proteolysis as receptor activation. To understand the mechanisms underlying how the frictional shear stress exerted by flow stimulates Notch1 receptor activation, we first tested a requirement for Notch ligands. Soluble recombinant DLL4 ECD (rhDLL4) binds to Notch1 to competitively inhibit ligand binding (Scehnet et al, 2007), but does not stimulate Notch1 proteolytic

activation, ultimately suppressing ligand-mediated receptor activation (Fig S1B and C). Treatment of hdBECs with rhDLL4 reduced basal Notch1 activation and prevented flow-induced increase in Notch1 activation (Fig 1A and B), indicating a requirement for Notch ligands. We next used CRISPR/Cas9 (Polacheck et al, 2017; Mayo et al, 2025 *Preprint*) to ablate the ligands DLL4 (*DLL4$^{KO}$*) or Jagged 1 (*JAG1$^{KO}$*) in hdBECs. Relative to a nontargeting scramble control (SCR), deletion of Jag1 increased both basal and flow-stimulated Notch1 activation. In contrast, deletion of DLL4 prevented Notch1 proteolytic activation in response to flow (Fig 1C and D), identifying an essential role of DLL4 in flow-mediated Notch1 activation.

To test whether Notch1 proteolytic activation by flow occurs via direct mechanosensing by the Notch1-DLL4 complex, that is, expression of DLL4 and Notch1 is sufficient to confer Notch1 receptor proteolysis by flow, we examined flow response in another endothelial subtype, primary human dermal lymphatic endothelial cells (hdLECs), which express comparable levels of Notch1, DLL4, Jag1, and VE-cadherin to hdBECs. However, in contrast to hdBECs, application of the identical magnitude of unidirectional laminar shear stress to hdLECs does not result in increased Notch1 proteolytic activation (Fig 1E and F), suggesting a distinct mechanotransductive process. To investigate potential mechanisms underlying this differential response, we examined hdLEC and hdBEC remodeling in response to flow by microscopy. Randomly oriented hdBECs and hdLECs under static conditions reorient to align along the axis of flow (Fig 1G). Notch1 is distributed across the plasma membrane and nucleus in both hdBECs and hdLECs cultured under static conditions (Fig 1G). Interestingly, in agreement with previous observations in human aortic endothelial cells (Mack et al, 2017), Notch1 robustly polarized to domains located at the downstream end of hdBECs, but this polarization was absent in hdLECs (Fig 1G and H). Time-lapse imaging of hdBEC monolayers mosaically expressing GFP-tagged Notch1 (Notch1-GFP) revealed that Notch1 polarization occurs within minutes of flow initiation and, importantly, that these polarized domains are the product of receptor translocation within the cell bearing the tagged receptor (Fig 1I; Video 1). Together, these data suggest that Notch1 polarization and proteolytic activation are products of a specific shear stress mechanotransduction pathway operating in hdBECs, but not hdLECs.

Curiously, quantification of Notch1 subcellular localization in *DLL4$^{KO}$* compared with SCR hdBECs revealed no difference in downstream Notch1 polarization in response to flow (Fig 1J and K). We then investigated whether DLL4 or Jag1 exhibited changes in subcellular localization in response to flow. To facilitate visualization of either ligand, we expressed fluorescently tagged DLL4 (DLL4-GFP) or Jag1 (Jag1-mEmerald) in hdBECs. The expression of either ligand was competent to increase basal Notch1 activation levels, yet the localization of DLL4 or Jag1 did not noticeably polarize in response to flow (Figs 1L and S1D–F). Altogether, although flow-induced Notch1 proteolytic activation specifically requires expression of and binding to DLL4, DLL4 does not control flow-induced downstream polarization. We therefore sought to understand the relationship between flow-mediated Notch1 polarization and receptor proteolytic activation.

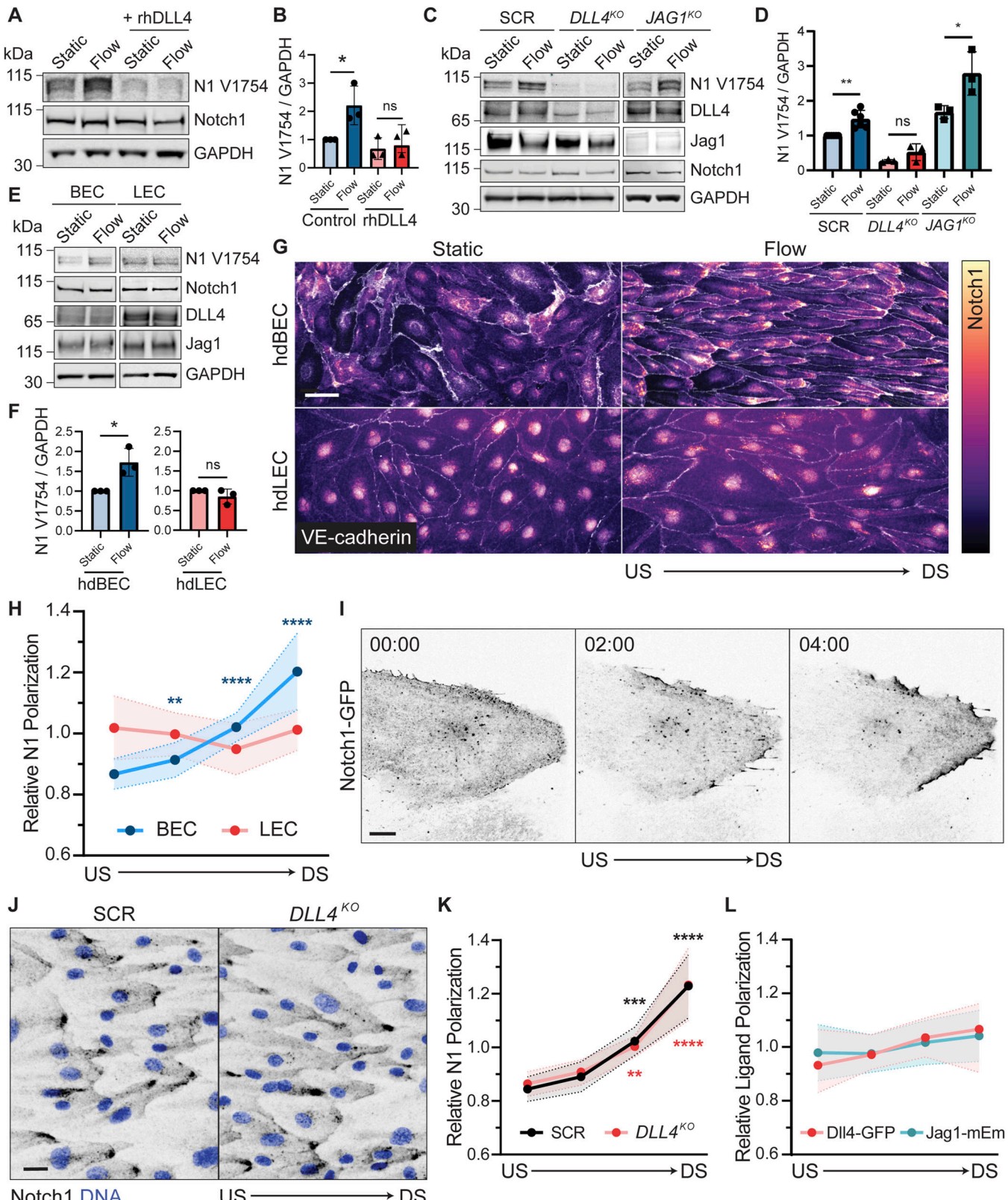

**Figure 1. DLL4 is required, but not sufficient, for Notch1 activation by shear stress.**
**(A)** Western blot of hdBEC lysates cultured under static or flow (~20 dyn/cm$^2$) conditions for 24 h, with pretreatment with 0.5 μg/ml of rhDLL4 or vehicle control. **(B)** Quantification of normalized Western blot intensity of Notch1 V1754. **(C)** Western blot of hdBEC lysates from scramble (SCR), *DLL4$^{KO}$*, and *JAG1$^{KO}$* hdBECs under static and flow conditions. **(D)** Quantification of normalized Western blot intensity of Notch1 V1754. **(E)** Western blot of hdBEC and hdLEC lysates under static and flow conditions. **(F)** Quantification of normalized Western blot intensity of Notch1 V1754. **(G)** Fluorescence micrographs of hdBECs and hdLECs under static and flow conditions

## Notch1 is cis-endocytosed in flow-polarized domains

We next characterized the nature and precise subcellular positioning of Notch1 within flow-polarized domains. We used mosaic hdBEC monolayers, composed of cells expressing either cytoplasmic mApple or mEmerald, cultured under flow to visualize cell–cell interfaces. In a representative monolayer interface under flow, a mApple-expressing cell is positioned upstream of a mEmerald-expressing cell and Notch1 polarizes to the downstream end of the mApple-expressing cell (Fig 2A). Interestingly, Notch1 accumulation initiates at a discrete apical cell–cell interface between the two cells (Fig 2A) but is not present in basal overlapping regions of the cell membrane. Full-length Notch1 is a noncovalent heterodimer at the plasma membrane, and immunostaining with antibodies recognizing both the Notch1 ECD and ICD reveals polarization within downstream domains (Figs 1 and 2A), suggesting polarization is an accumulation of the full-length receptor and not a product of proteolytic cleavage. To further test the hypothesis that full-length Notch1 heterodimers polarize downstream of flow, we treated hdBECs with either BB94 (Batimastat), a metalloprotease inhibitor that prevents cleavage of Notch1 at the S2 site to release the ECD (Hayward et al, 2019), or N-[N-(3,5-difluorophenacetyl)-L-alanyl]-S-phenylglycine t-butyl ester (DAPT), an inhibitor of γ-secretase–mediated cleavage of Notch1 at the S3 site to release the ICD (Hellström et al, 2007). Treatment with BB94 or DAPT did not prevent Notch1 polarization under shear stress, confirming that full-length Notch1 receptors polarize in response to flow (Fig 2B and C).

Canonical activation of Notch receptors occurs by ligand trans-endocytosis, in which the Notch ECD is endocytosed by the ligand-expressing cell allowing ICD release in the opposing cell (Gordon et al, 2015; Sprinzak & Blacklow, 2021). Under this model, Notch1 ECD and ICD would be predicted to translocate in opposing directions after proteolytic activation. Interestingly, quantification of ECD and ICD distribution within flow-polarized domains at cell–cell interfaces revealed overlapping polarized regions of ECD and ICD within the upstream cell, with no evident ECD trans-endocytosis or ICD in the adjacent downstream cell (Fig 2D and E). This suggested a mechanism distinct from trans-endocytosis is operating in flow-polarized domains.

We next investigated Notch1 receptor dynamics within flow-polarized domains. To visualize the dynamics of endogenous Notch1 receptors, we incubated hdBECs with a fluorescent dye-conjugated, monoclonal antibody directed toward the Notch1 ECD while under flow (Kuintzle et al, 2025). Incubation with the conjugated antibody robustly labeled flow-polarized Notch1 domains across the monolayer (Fig 2F, Video 2). Consistent with a mechanism distinct from trans-endocytosis, time-lapse imaging revealed

rapid endocytosis of labeled Notch1 receptors within flow-polarized domains and their retrograde transport toward the cell interior (Fig 2F; Video 2). No labeled Notch1 was observed to cross the cell–cell interface and enter the opposing cell. We then adapted this assay to antibody pulse labeling, allowing quantification of internalized Notch1 over a defined period. Pulse labeling of SCR control compared with *DLL4^{KO}* cells revealed no difference in Notch1 internalization, indicating that DLL4 does not control Notch1 cis-endocytosis (Fig 2G–I). Therefore, flow-polarized domains contain full-length Notch1 heterodimers that are dynamically cis-endocytosed independent of DLL4 ligation.

## Notch1 ICD is required for flow-induced polarization and proteolytic activation

Notch1 polarization and endocytosis within flow-polarized domains are independent of DLL4. We therefore next asked whether these processes are governed by domain-specific functions inherent to the Notch1 receptor. Our previous work identified the Notch1 transmembrane domain (TMD) is sufficient to target to endothelial adherens junctions (Polacheck et al, 2017). The expression of a SNAP-tagged TMD in flow-aligned hdBECs indeed revealed localization to cell–cell adhesions, but not to polarized downstream domains (Fig S1G). We next investigated whether a domain-specific polarization function exists for the ICD.

Previous studies have demonstrated that replacing mouse Notch1 ICD at Arg1752 with a nuclear-targeting Cre recombinase is sufficient to trace Notch1 proteolytic activation in vivo (Vooijs et al, 2007). Adopting this approach, we engineered hdBECs using a knockdown-rescue system, which delivers a shRNA targeting the *NOTCH1* 3′ untranslated region concurrent with the expression of either a C-terminal mEmerald-tagged WT Notch1 (Notch1-WT) or a truncated variant in which the Notch1 ICD has been replaced with mEmerald at the human equivalent residue Arg1762 (Notch1-ΔICD) (Fig 3A). In hdBECs, both Notch1-WT and Notch1-ΔICD are expressed at the cell surface (Fig S1H). Application of flow to cells expressing Notch1-WT resulted in increased Notch1 proteolytic activation, evidenced by the cleavage-specific Notch1 V1754 antibody at a larger molecular weight because of the addition of mEmerald (Fig 3B). Furthermore, culturing cells expressing Notch1-WT on substrates coated with immobilized rhDLL4, which increases basal Notch1 proteolytic activation in hdBECs (Fig S1I), similarly results in increased activation (Fig 3B). In contrast, activation of Notch1-ΔICD did not increase in response to flow and is diminished when plated on immobilized rhDLL4 substrates (Fig 3C). However, Notch1-ΔICD is indeed sensitive to canonical ligand proteolytic transactivation, as mosaic coculture of Notch1-ΔICD with increasing proportions of hdBECs overexpressing DLL4-mScarlet leads to proportional

---

immunostained for Notch1 intracellular domain (heatmap) and VE-cadherin (white). Scale bar, 25 $\mu$m. **(H)** Quantification of the relative Notch1 polarization in hdBECs versus hdLECs under flow. $n \geq 10$ fields of view from three independent experiments. Statistical significance of quadrants of each condition relative to an unpolarized line is indicated. **(I)** Time lapse of Notch1-GFP in hdBECs under flow. Timescale, minute:second. Scale bar, 10 $\mu$m. **(J)** Fluorescence micrographs of SCR and *DLL4^{KO}* cells under flow conditions immunostained for Notch1 (black) and DNA (blue). Scale bar, 25 $\mu$m. **(K)** Quantification of the relative degree of Notch1 polarization in SCR versus *DLL4^{KO}* cells under flow. $n \geq 10$ fields of view from three independent experiments. **(L)** Quantification of the relative degree of ligand polarization in DLL4-GFP or Jag1-mEmerald cells under flow. $n \geq 10$ fields of view from three independent experiments. Statistical significance of quadrants of each condition relative to an unpolarized line is indicated. Western blots are representative of three independent experiments. US, upstream; DS, downstream. For all plots, mean ± SD; one-way ANOVA with Tukey's post hoc test, *$P < 0.05$, **$P < 0.01$, ***$P < 0.001$, ****$P < 0.0001$, ns denotes nonsignificant.
Source data are available for this figure.

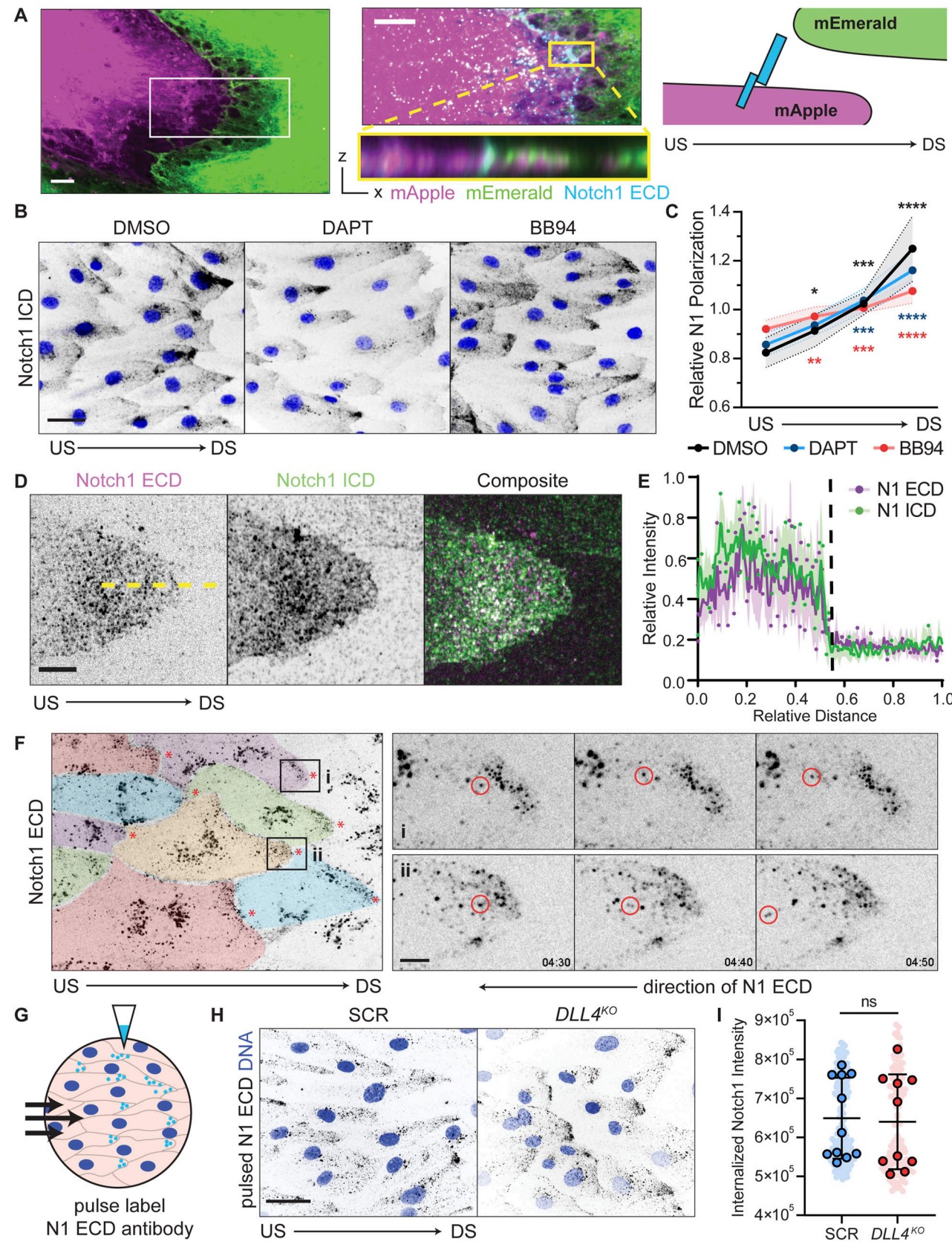

increases in Notch1-ΔICD proteolysis (Fig 3D). Consistent with the differential effects of flow on Notch1-WT or Notch1-ΔICD activation and the central role of receptor polarization in flow-mediated activation, Notch1-WT, but not Notch1-ΔICD, polarizes to downstream domains in response to flow (Fig 3E and F).

Altogether, these results support a model in which the Notch1 ICD is critical for downstream flow polarization of the receptor. Loss of the Notch1 ICD prevents increased receptor activation in response to flow and when cultured on immobilized rhDLL4 substrate but retains increased activation in response to DLL4 sender–receiver transactivation. This suggests distinct regulatory mechanisms exist to control Notch1 activation in specific cellular contexts. We thus sought to identify mechanisms operating on the ICD in response to flow.

### ICD associations with annexin A2 and caveolin-1 regulate receptor endocytosis and proteolytic activation in response to flow

To identify putative flow-induced ICD interactions, we unbiasedly profiled interacting proteins from hdBECs cultured under static or flow conditions using immunoprecipitation of the Notch1 ICD, SDS–PAGE and Coomassie staining, and band-excision mass spectrometry (Figs 4A and B and S2A and B, Table S1). We focused on interactions showing unique or elevated abundance in co-immunoprecipitation under flow (Fig 4A and B). One prominent interaction identified was annexin A2, a calcium- and phospholipid-binding protein that regulates membrane-associated actin dynamics and endo/exocytic pathways (Grieve et al, 2012; Bharadwaj et al, 2013; Luo & Hajjar, 2013), and was reported as an abundant Notch1 ICD interactor in a recent proteomic study (Bian et al, 2023). Co-immunoprecipitation and Western blot confirmed increased annexin A2 and Notch1 interaction in hdBECs under flow (Fig 4B), and immunofluorescence staining reveals annexin A2 accumulates at both adherens junctions and flow-polarized downstream domains (Fig 4E). CRISPR-mediated deletion of annexin A2 ($ANXA2^{KO}$) reduced Notch1 proteolytic activation, but not total Notch1 protein levels, in hdBECs under flow (Fig 4C and D). Although $ANXA2^{KO}$ does not prevent Notch1 polarization in response to flow (Figs 4H and S2C), pulse labeling revealed significantly reduced Notch1 internalization upon loss of annexin A2 (Fig 4F and G). Altogether, these results identify annexin A2 is a novel regulator of Notch1 activation by flow and connect a function of annexin A2 in receptor cis-endocytosis to Notch1 proteolytic activation.

A second prominent group of interactions identified via mass spectrometry included several caveolar proteins including cavins and EH domain–containing proteins (Fig 4A). Caveolae are mechanically sensitive membrane domains that have been reported to polarize similar to Notch1 in response to flow, defining the membrane mechanics of flow-polarized domains and playing critical anti-inflammatory roles in endothelia (Del Pozo et al, 2021; Hong et al, 2024). Co-immunoprecipitation of Notch1 ICD and caveolin-1 increases under flow (Fig 4B), and given that several caveolar proteins were identified to interact with ICD, we sought to globally ablate caveolae via CRISPR/Cas9-mediated deletion of caveolin-1 ($CAV1^{KO}$), deletion of which has been demonstrated to deplete caveolae (Drab et al, 2001; Gu et al, 2014). Immunofluorescence staining confirmed that caveolin-1 polarizes to downstream domains in response to flow (Fig 4I). $CAV1^{KO}$ in hdBECs prevented increased Notch1 proteolytic activation in response to shear stress and to immobilized rhDLL4 substrates, but did not affect total Notch1 protein levels (Fig 4J and K). Loss of caveolin-1 did not impair Notch1 polarization in response to flow, and, reciprocally, loss of Notch1 did not prevent caveolin-1 polarization (Figs 4M and S3A and B). Furthermore, pulse labeling revealed $CAV1^{KO}$ did not prevent Notch1 flow-induced internalization (Figs 4L and S3C). However, consistent with a central role in Notch1 shear stress mechanotransduction, immunofluorescence staining revealed polarized caveolin-1 microdomains tightly colocalize with Notch1 in hdBECs, but do not form in hdLECs (Fig 4N). These data support a model in which the polarization of Notch1 and caveolin-1 functions to enhance Notch1 proteolytic activation in response to fluid shear stress.

## Discussion

The goal of this study was to identify the mechanisms linking the physical forces of hemodynamic shear stress to the proteolytic activation of endothelial Notch1 signaling. We determine that Notch1 receptor cleavage and ICD release in response to laminar shear stress are a product of a mechanotransduction cascade operating in blood microvascular endothelia, but not tissue-matched lymphatic endothelia. Although DLL4 is required for this process, DLL4 expression is not sufficient for Notch1 shear stress mechanotransduction. Notch1 activation involves translocation of the receptor to flow-polarized domains in the plasma membrane, where live-cell imaging revealed the receptor is rapidly

---

**Figure 2. Notch1 is cis-endocytosed in flow-polarized domains.**
**(A)** Left: fluorescence micrographs of coculture of hdBECs expressing mEmerald (green) or mApple (magenta) under flow. Scale bar, 5 μm. Inset: high-magnification micrograph of the cell–cell interface with mApple, mEmerald, and Notch1 extracellular domain (ECD) (cyan). Yellow inset: XZ orthogonal projection. Scale bar, 5 μm. Right: experimental schematic. **(B)** Fluorescence micrographs of hdBECs pretreated for 1 h with 10 μM DAPT, 1 μM BB94, or DMSO vehicle control under flow. Scale bar, 25 μm. **(C)** Quantification of Notch1 polarization in hdBECs treated with DAPT, BB94, or DMSO vehicle control under flow. n ≥ 10 fields of view from three independent experiments. Statistical significance of quadrants of each condition relative to an unpolarized line is indicated. **(D)** Super-resolution by optical pixel reassignment fluorescence micrographs of the downstream cell–cell interface of flow-polarized hdBECs. Scale bar, 5 μm. **(E)** Line-scan quantification of relative Notch1 ECD (magenta) and Notch1 intracellular domain (green) distribution (representative yellow dashed line). n = 10 profiles. **(F)** Fluorescence time series from live-cell movie of hdBECs labeled with AF647-Notch1 ECD antibody under flow. Pseudocolored to depict cell–cell boundaries. Insets (i) and (ii) of distinct polarized domains where individual particles (red circles) are tracked over time moving retrograde opposite the direction of flow. Timescale, minute:second. Scale bar, 5 μm. **(G)** Schematic illustrating pulse Notch1 ECD antibody labeling experiments. **(H)** Fluorescence micrographs of internalized Notch1 in SCR and $DLL4^{KO}$ cells under flow conditions from pulse labeling of Notch1 ECD (black). Scale bar, 25 μm. **(I)** Quantification of endocytosed Notch1 in SCR versus $DLL4^{KO}$ cells. n ≥ 10 fields of view from three independent experiments. US, upstream; DS, downstream. For all plots, mean ± SD; one-way ANOVA with Tukey's post hoc test, *P < 0.05, **P < 0.01, ***P < 0.001, ****P < 0.0001, ns denotes nonsignificant.

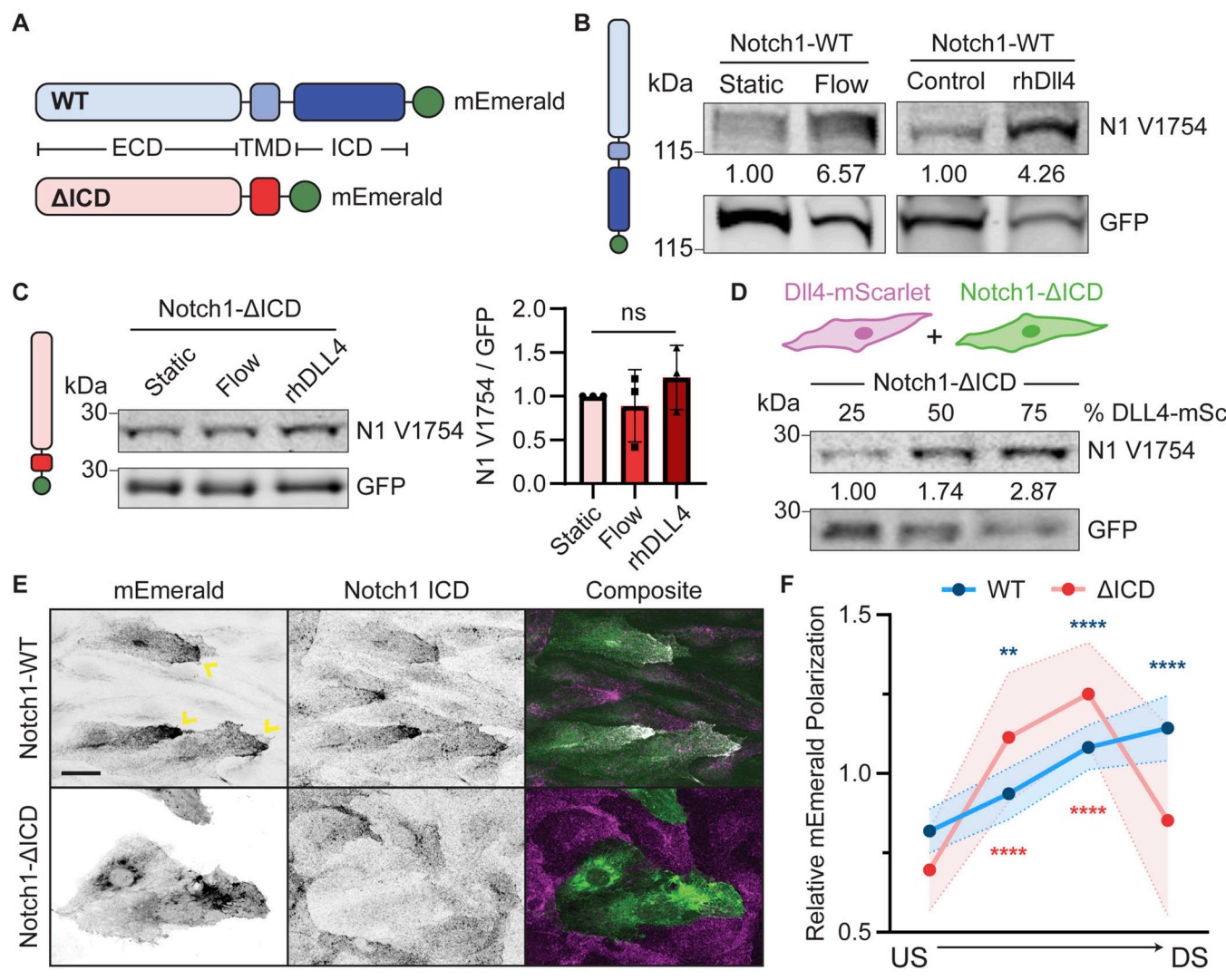

**Figure 3. Notch1 intracellular domain (ICD) is required for flow-induced polarization and proteolytic activation.**
**(A)** WT Notch1 (WT) or a Notch1 mutant lacking the intracellular domain (ΔICD) C-terminally tagged with mEmerald. **(B)** Western blots of Notch1-WT–expressing hdBEC lysates cultured under static and flow conditions or plated on control or rhDLL4-coated substrate. Quantification: Notch1 V1754 intensity normalized to total construct expression (mEmerald recognized by GFP antibody). **(C)** Western blot of Notch1-ΔICD–expressing hdBEC lysates cultured under static and flow conditions or plated on rhDLL4-coated dishes. Quantification: Notch1 V1754 intensity normalized to total construct expression (mEmerald recognized by GFP antibody). Quantification of Western blot intensity of Notch1 V1754 normalized to GFP. $n = 3$ independent experiments. **(D)** Western blot of lysates from Notch1-ΔICD–expressing hdBEC monolayers mosaically cocultured with increasing percentage of DLL4-mScarlet–overexpressing hdBECs. Quantification is Notch1 V1754 intensity normalized to GFP. **(E)** Fluorescence micrographs of Notch1-WT– and Notch1-ΔICD–expressing hdBECs under flow. Yellow arrows denote flow-polarized domains. Scale bar, 25 $\mu$m. **(F)** Quantification of Notch1 polarization in Notch1-WT– and Notch1-ΔICD–expressing hdBECs under flow. $n \geq 10$ fields of view from three independent experiments. Statistical significance of quadrants of each condition relative to an unpolarized line is indicated. US, upstream; DS, downstream. For all plots, mean ± SD; one-way ANOVA with Tukey's post hoc test, **$P < 0.01$, ****$P < 0.0001$, ns denotes nonsignificant.
Source data are available for this figure.

cis-endocytosed. Searching for domain-specific Notch1 functions in this process, we investigated the role of the ICD using a truncated form of the receptor. Although truncation of ICD permits classic ligand–receptor transactivation, it prevents Notch1 polarization and proteolytic activation in response to flow. Using mass spectrometry, we identify that flow stimulates ICD interaction with annexin A2, which controls receptor activation via cis-endocytosis, and caveolin-1 rich domains, which spatially compartmentalize with Notch1 and contribute to receptor proteolytic activation. Altogether, this work identifies new mechanisms regulating Notch receptor activation

that operate through the ICD during fluid shear stress mechanotransduction (Fig 4O).

A central finding of this work is that Notch1 mechanotransduction in response to shear stress occurs through cis-endocytosis of the receptor, a process that governs flow-sensitive molecular interactions with the ICD. We further show that the ICD is required for Notch1 activation by immobilized rhDLL4, suggesting a broader regulatory roles of the ICD beyond flow sensing. Notch cis-endocytosis has previously been shown to regulate receptor activation in *Drosophila* S2 cells through Deltex E3 ubiquitin

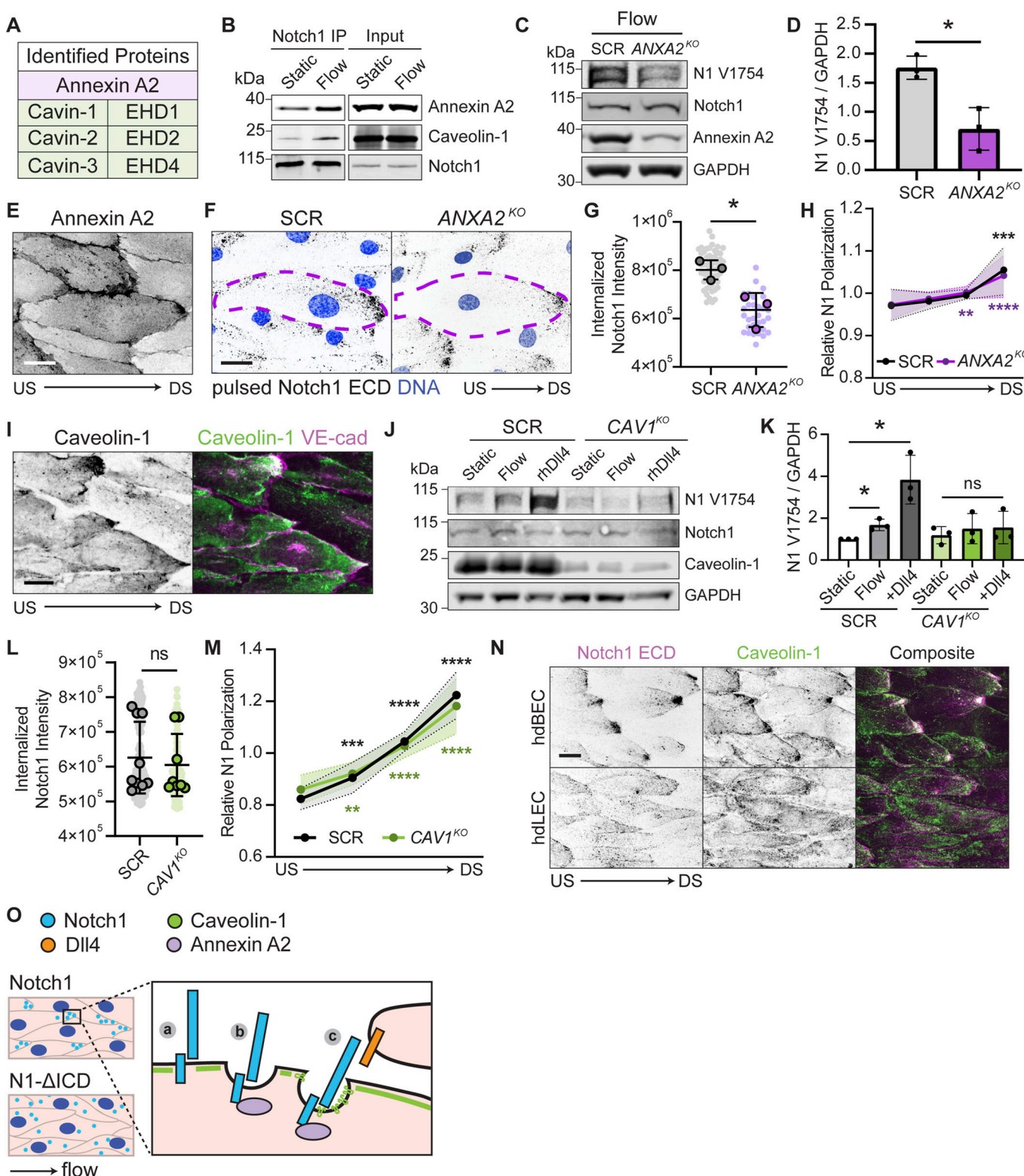

**Figure 4. Intracellular domain (ICD) associations with annexin A2 and caveolin-1 regulate receptor endocytosis and proteolytic activation in response to flow.**
**(A)** Select list of ICD-interacting proteins identified via mass spectrometry to increase under flow. **(B)** Western blot of Notch1 co-immunoprecipitation from hdBECs cultured under static or flow conditions. **(C)** Western blot of lysates from SCR and *ANXA2^KO^* hdBECs under flow. **(D)** Quantification of Western blot intensity of Notch1 V1754 normalized to GAPDH. Data are normalized to SCR for *n* = 3 independent experiments. **(E)** Fluorescence micrograph of annexin A2 in hdBECs under flow. Scale bar, 25 *μm*. **(F)** Fluorescence micrographs of endocytosis of Notch1 from pulse labeling of SCR and *ANXA2^KO^* cells. Purple dashed lines indicate cell segmentation. Scale bar, 25 *μm*. **(G)** Quantification of internalized Notch1 extracellular domain in SCR versus *ANXA2^KO^* cells. *n* ≥ 10 fields of view from three independent experiments. **(H)** Associated quantification of the relative degree of Notch1 polarization in SCR versus *ANXA2^KO^* cells under flow, measured as described previously. *n* ≥ 10 fields of view

ligase–mediated ICD ubiquitination (Chastagner et al, 2017). Notably, although DLL4 is required for flow-mediated Notch1 activation, loss of DLL4 does not affect Notch1 polarization or endocytosis. These findings support a model in which Notch1 receptors undergo continuous cis-endocytosis while polarized within downstream microdomains, and in which engagement with DLL4 at the apical cell–cell interface before endocytosis enables proteolytic activation. In this context, the precise role of ADAM-mediated S2 cleavage remains unresolved, as it may occur either extracellularly or within endosomal compartments (Caolo et al, 2020; Shimizu et al, 2024). In addition, the accumulation of Notch1 puncta observed by antibody pulse labeling suggests that receptor clustering at the plasma membrane may be a prerequisite for cis-endocytosis. Consistent with this idea, caveolin-1 binding to the ICD regulates Notch1 activation in response to flow and immobilized DLL4, although the underlying mechanism remains unclear. Caveolin-1 may promote Notch1 clustering within polarized microdomains or alternatively regulate its compartmentalization with γ-secretase, mechanisms that may explain aberrant Notch signaling in cerebral cavernous malformations associated with *PDCD10* deletion, where caveolar biogenesis is increased (You et al, 2013; Zhou et al, 2021).

We demonstrate that Notch1 translocation to flow-polarized domains within the plasma membrane is a critical step of receptor proteolytic activation. Notch1 polarization by flow occurs independently of DLL4 and requires the Notch1 ICD, but the ultimate mechanisms directing receptor translocation remain undetermined. A hint may lie in the speed of polarization in response to flow and evidence in assorted systems for noncanonical Notch functions related to actin (Ben-Yaacov et al, 2001; Major & Irvine, 2005, 2006; Langdon et al, 2006; Wesley et al, 2011; Zhang & Yang, 2016; Handa et al, 2022; White et al, 2023). Indeed, endothelial cortical actin remodeling occurs rapidly at the onset of flow (Galbraith et al, 1998; Choi & Helmke, 2008; Mengistu et al, 2011). Given the established interaction between cortical actin and membrane constituents like phosphatidylserine that stabilize flotillin-containing microdomains (Kwak et al, 2022; Sych et al, 2022) and that TMD alone does not flow-polarize, we speculate that a coupling of the ICD (either directly or indirectly) to cortical actin dynamics may underlie receptor translocation.

Notch1 is part of a growing list of proteins that flow discretely polarizes to downstream microdomains in arterial and microcirculation-derived endothelia, including caveolin-1, VE-PTP, and heart of glass (HEG1) (Baluk et al, 2023; Hong et al, 2024; Tamargo et al, 2024). These membrane microdomains exist in vivo, are enriched in raft-associated lipids, and have increased membrane tension. These domains play key roles in determining endothelial phenotype, including calcium entry, regulation of inflammatory gene expression, and Tie2 receptor signaling. Here, we

determine how these microdomains function as a regulatory "hotspot" for Notch1 signaling. Given the apparent endothelial subtype specificity of Notch1 flow mechanotransduction, this may serve to reinforce flow-mediated endothelial transcriptional programs in arteries and microcirculation. Still, the connection of these microdomains to major endothelial signaling programs warrants deeper proteomic profiling and a greater understanding of what endothelial mechanosensory mechanisms exist to assemble these signaling domains with speed and spatial precision.

Altogether, this study identifies new mechanisms regulating Notch1 shear stress mechanotransduction in endothelial cells that should improve our understanding of Notch regulation and function in angiogenesis, barrier function regulation, and other Notch-associated vasculopathies. More broadly, we establish the importance of Notch ICD in controlling receptor localization and a new context in which the ICD facilitates protein–protein interactions necessary for proteolytic activation.

# Materials and Methods

## Cell culture

Primary sorted human dermal microvascular blood endothelial cells (hdBECs, 0% Prox1+; Lonza) and human dermal lymphatic cells (hdLECs, 99% Prox1+; Lonza) were maintained in EGM-2MV growth media (Lonza) from passages 2–8 and passaged at 80–90% confluency. End-point experiments (e.g., biochemical analysis or fixation for immunofluorescence) were cultured until cells were fully confluent. HEK-293T cells (Clontech) were maintained from passages 3–20 in high-glucose DMEM (Sigma-Aldrich) supplemented with 10% FBS (Peak), 2 mM L-glutamine (Invitrogen), 1 mM sodium pyruvate (Gibco), and 1% penicillin/streptomycin (Sigma-Aldrich). Cells were cultured in a 37°C humidified incubator with 5% $CO_2$. Cell-line authentication (performance, differentiation, and STR profiling) was provided by Lonza. Cells were routinely tested for *Mycoplasma* via the PCR test (Applied Biological Materials).

## Antibodies and reagents

Antibodies against Notch1 V1754 (V1744 in mice, D3B8, 1:500; WB), Notch1 ICD (D1E11, 1:200 IF, 1:1,000; WB), GAPDH (14C10, 1:5,000; WB), DLL4 (D7N3H, 1:1,000; WB), Jag1 (D4Y1R, 1:1,000; WB), GFP (D5.1, 1:5,000; WB), annexin A2 (D11G2, 1:1,000; WB, 1:200; IF), presenilin-1 (E3L9X, 1:1,000; WB), and flotillin-2 (C42A3, 1:1,000; WB) were from Cell Signaling Technologies. Notch1 ECD (MHN1-519, 1:200; IF) was from BD Biosciences. Antibodies against caveolin-1 (PA1-064, 1:

from three independent experiments. Statistical significance of quadrants of each condition relative to an unpolarized line is indicated. **(I)** Fluorescence micrographs of caveolin-1 and VE-cadherin in hdBECs under flow. Scale bar, 25 $\mu$m. **(J)** Western blot of hdBEC lysates from SCR and *CAV1^{KO}* cells cultured under static and flow conditions or plated on rhDLL4-coated substrates. **(K)** Quantification of Western blot intensity of Notch1 V1754 normalized to GAPDH. Data are normalized to SCR static condition. **(L)** Quantification of internalized Notch1 extracellular domain by pulse labeling in SCR versus *CAV1^{KO}* cells. n ≥ 10 fields of view from three independent experiments. **(M)** Quantification of Notch1 polarization in SCR versus *CAV1^{KO}* cells under flow. n ≥ 10 fields of view from three independent experiments. Statistical significance of quadrants of each condition relative to an unpolarized line is indicated. **(N)** Fluorescence micrographs of hdBECs and hdLECs under flow. Scale bar, 25 $\mu$m. **(O)** Schematic depicting flow-polarized downstream domains where (a) ICD localizes full-length Notch1, (b) annexin A2 regulates Notch1 cis-endocytosis, and (c) caveolin-1 contributes to Notch1 proteolysis through a to-be-determined mechanism. US, upstream; DS, downstream. For all plots, mean ± SD; one-way ANOVA with Tukey's post hoc test, *$P < 0.05$, **$P < 0.01$, ***$P < 0.001$, ****$P < 0.0001$, ns denotes nonsignificant. Source data are available for this figure.

2,500; WB, 1:200; IF) and ZO-1 (R26.4C, 1:200; IF) were from Thermo Fisher Scientific. Antibodies against VE-cadherin (F-8, 1:100; IF) and Notch1 ICD (A-8, 1:300; WB) were from Santa Cruz. Rhodamine phalloidin, and Alexa Fluor 488, 568, and 647 goat anti-mouse and anti-rabbit IgG secondary antibodies (1:400) were from Invitrogen. DAPI nuclear stain was from Sigma-Aldrich. Anti-SNAP (P9310S, 1:1,000; WB) was from New England Biolabs. IRDye donkey anti-rabbit and anti-mouse IgG secondary antibodies (1:5,000) were from LI-COR. DAPT (10 $\mu$M) was from Sigma-Aldrich. BB-94 (1 $\mu$M) was from Selleckchem. rhDLL4 (0.5 $\mu$g/ml) and rhJag1 (1.0 $\mu$g/ml) were from R&D Systems.

### Lentiviral-mediated CRISPR/Cas9 editing

Stable CRISPR-modified primary hdBEC lines were generated using our previously established protocols (Polacheck et al, 2017; White et al, 2023; Mayo et al, 2025 Preprint). Specific guide RNAs were cloned into the BsmBI site of plentiCRISPRv2: SCR, 5′-GTATTACTG ATATTGGTGGG-3′, DLL4[KO], 5′-CATCAACGAGCGCGGCGTAC-3′, JAG1[KO], 5′-CGCGGGACTGATACTCCTTG-3′, ANXA2[KO], 5′-ACAGGGGCTGGGAAC CGACG-3′, CAV1[KO], 5′-AGTGTACGACGCGCACACCA-3′, and NOTCH1[KO], 5′-CGTCAGCGTGAGCAGGTCGC-3′. sgRNA-containing plentiCRISPRv2 plasmids were cotransfected with psPAX2 (plasmid #12260; Addgene) and pMD2.G (plasmid #12259; Addgene) packaging plasmids into HEK-293T cells using calcium phosphate transfection. After 48 h, viral supernatants were collected from the culture dish and passed through a 0.22-$\mu$m sterile filter. Lentivirus was incubated with 1X polyethylene glycol lentiviral concentrator solution (PEG-IT, 3X solution: 10% wt/vol polyethylene glycol 8,000 and 0.3 M NaCl) overnight on a rotator at 4°C, then spun down at 1,600$g$ at 4°C for 1 h. The viral pellet was resuspended in sterile PBS, then stored in −80°C. Approximately 200 k hdBECs were transduced with 50–100 $\mu$l of lentivirus in EGM2-MV media overnight and given fresh media the next morning. At 48 h post-transduction, cells were batch-selected with 2 $\mu$g/ml puromycin for 2 d. All CRISPR modifications were verified by Western blot.

### Lentiviral-mediated gene expression

Full-length human Notch1 cDNA was a gift from Michael Elowitz. Human DLL4 and Jag1 were cloned from hdBEC cDNA libraries. All fusion proteins were assembled into a lentiviral pRRL vector at the KpnI and EcoRI sites and expressed using a CMV promoter. DNA fragments were assembled using NEBuilder HiFi DNA Assembly Master Mix. For knockdown-rescue constructs, a shRNA targeting the 3′ UTR of NOTCH1 (GGAAACAAGTGAAAGCATA) was cloned into the HpaI and XhoI sites of pLL3.7 (plasmid #11795; Addgene). Either full-length Notch1 (Notch1-WT) or Notch1 truncated at Arg1762 (Notch1-ΔICD) fused on the C terminus with mEmerald (plasmid # 54025; Addgene) was cloned after the CMV promoter in pLL3.7 using AgeI and EcoRI. Notch1, DLL4, and Jag1 cDNA sequences were verified by whole-plasmid sequencing. All lentiviral-based plasmids, except the knockdown-rescue constructs, were transfected into HEK-293T cells. To improve transfection efficiency of the knockdown-rescue constructs into HEK-293Ts, the knockdown-rescue plasmids were cotransfected with psPAX2 (plasmid #12260; Addgene) and pMD2.G (plasmid #12259; Addgene)

packaging plasmids into HEK-293T cells using jetOPTIMUS, a cationic lipid transfection reagent (Polyplus). Lentivirus was incubated with 1X polyethylene glycol lentiviral concentrator solution (PEG-IT, 3X solution: 10% wt/vol polyethylene glycol 8,000 and 0.3 M NaCl) overnight on a rotator at 4°C, then spun down at 1,600$g$ at 4°C for 1 h. The viral pellet was resuspended in sterile PBS, then stored in −80°C.

### Recombinant ligand coating and flow experiments

To expose cells to immobilized ligand, six-well tissue culture plates were surface-activated by plasma treatment for 30 s and coated with recombinant ligand (0.5 $\mu$g/ml rhDLL4, 1.0 $\mu$g/ml rhJag1; R&D Systems) suspended in sterile PBS for 1 h at 37°C on a rocker (Harrington et al, 2008).

For live-cell confocal imaging, cells were seeded into glass-bottom parallel plate flow chambers ($\mu$-Slide I$^{0.2}$ Luer, ibidi) such that they reached full confluence the next morning and a peristaltic pump (Avantor) was used to apply 20 dyn/cm$^2$ of shear stress. For application of shear stress for fixed confocal imaging, 250 k cells were first plated on 25-mm glass coverslips, which were surface-activated by plasma treatment for 30 s, coated with 50 $\mu$g/ml collagen type I (Corning) in PBS (Gibco) for 1 h at 37°C, and washed three times with PBS. Once prepared, coverslips were placed within six-well plates (VWR) and cultured for 2 d until confluent. For biochemical analysis, 250 k cells were plated directly into six-well plates and cultured for 2 d until confluent. In each case, confluent monolayers in wells containing 2 ml of media were exposed to shear stress for 1 or 24 h on an orbital shaker (VWR Model 3500) at 37°C at a rotation speed of 400 RPM, ~20 dyn/cm$^2$ (Warboys et al, 2019).

### Immunoblotting

To measure protein expression, confluent hdBEC or hdLEC monolayers cultured in six-well plates were washed with PBS plus calcium and magnesium (PBS++, 0.9 mM CaCl$_2$, 0.49 mM MgCl$_2$) and lysed with cold 50 mM Tris–HCl, pH 7.4, 150 mM NaCl, 1% Triton X-100, 0.1% SDS, 0.5% sodium deoxycholate, and 1.5X Halt Protease and Phosphatase Inhibitor Cocktail (Thermo Fisher Scientific). Lysates were scraped and incubated on ice for 20 min before centrifugation at 4°C for 10 min at 15,000$g$. Lysate supernatants were collected, and protein content was normalized via BCA protein assay (Prometheus). Lysates were then denatured with 1X NuPAGE LDS Sample Buffer (Life Technologies) containing 5% $\beta$-mercaptoethanol for 10 min at 70°C. Denatured lysates were analyzed by SDS–PAGE, and gels were transferred to nitrocellulose membranes using Mini Trans-Blot Cell (Bio-Rad). Membranes were blocked in 5% nonfat dry milk in TBS containing 0.1% Tween-20 (TBST) for 1 h at room temperature (RT) and then incubated overnight at 4°C with primary antibodies in blocking buffer. Membranes were then washed with TBST three times over 30 min at RT. IRDye donkey anti-rabbit and anti-mouse IgG secondary antibodies were incubated in blocking buffer for 1 h at RT. Membranes were then washed with TBST three times over 30 min at RT. All immunoblots were imaged using Odyssey CLx LI-COR Imaging System and quantified using ImageJ. Immunoblots were adjusted

for brightness and contrast using ImageJ, and intensity values were normalized to the GAPDH loading control and biological replicate control (e.g., SCR Static).

### Immunofluorescence and live imaging

hdBEC and hdLEC monolayers were fixed in 4% paraformaldehyde (Electron Microscopy Sciences) in PBS++ for 10 min at 37°C and washed three times with PBS over 30 min at RT. Coverslips were then permeabilized in 0.1% Triton X-100 (Sigma-Aldrich) for 10 min at 37°C and washed three times over 30 min with PBS at RT. Next, coverslips were blocked in a blocking solution of 5% BSA and 4% goat serum in PBS for 1 h at RT. Primary antibodies were incubated overnight at 4°C or 1–2 h at RT in blocking buffer. Coverslips were then rinsed three times with PBS over 30 min. Secondary antibodies were incubated for 1 h at RT in blocking buffer. Coverslips were rinsed three times with PBS and then mounted to glass slides with mounting media (Invitrogen). Coverslips were sealed with clear nail polish (Electron Microscopy Science) and stored at 4°C. For fixed coverslips cultured under flow conditions, imaging was performed only at the periphery of the coverslip where cells were exposed to laminar flow.

For Notch1-ECD pulse labeling, 2.5 $\mu$l of Notch1 ECD antibody (BD Biosciences) was added to cells for the final 5 min during a 24-h flow application on the orbital shaker. Coverslips were washed twice with PBS++ before fixation and staining. For live imaging of Notch1-ECD, cells seeded into parallel plate flow chambers were flow-conditioned overnight. The next morning, cells were incubated with NOTCH1 ECD antibody conjugated to Alexa Fluor 647 (BD Biosciences) at 1:100 in medium for 30 min on a rocker at 37°C before imaging. For live imaging of Notch1-GFP, Notch1-GFP was transduced into cells at least 2 d before imaging.

Imaging was performed on the Yokogawa CSU-W1/SoRa spinning disk confocal system equipped with 405-, 488-, 561-, and 640-nm laser lines with a 20X 0.75 NA air objective (Nikon) or 60X 1.49 NA oil immersion lens (Nikon) and ORCA Fusion BT sCMOS camera (Hamamatsu) controlled through NIS-Elements software (Nikon) ([Dema et al, 2024](ref) Preprint). The imaging system was enclosed within an imaging chamber (In Vivo Scientific) at 37°C with humidified air at 5% $CO_2$. Fluorescence images were adjusted for contrast and brightness using ImageJ.

### Immunoprecipitation and mass spectrometry

hdBECs were cultured to confluence in six-well plates, three to four wells per condition, and cultured under static or flow conditions. Cells were washed with cold PBS++, lysed in cold IP lysis buffer (pH 7.4, 25 mM Tris, 150 mM NaCl, 5 mM $MgCl_2$, 1% vol/vol Triton X-100) with 2X Halt Protease and Phosphatase Inhibitor Cocktail (Thermo Fisher Scientific), scraped, and needle-lysed 12-15X using a 1-ml syringe and 22G needle. Lysates were rotated at 4°C for 30 min, then centrifuged for 10 min at 4°C at 15,000$g$. Supernatants were isolated from the pellet, flash-frozen with liquid $N_2$, and stored at −80°C if not used immediately. Protein concentration was normalized across all conditions using a BCA reaction kit (Prometheus). Before adding antibody, a small amount of lysate (25 $\mu$g) was set aside as "Input." For immunoprecipitation, primary

antibody was incubated on a tube rotator for 2 h at 4°C at a ratio of 1.5 $\mu$g antibody per 1 mg of protein in lysate. The antibody and lysate solution were incubated with 50 $\mu$l Pierce Protein A/G beads (Thermo Fisher Scientific) on a tube rotator for 2 h at 4°C. Beads were spun down and washed 3X with IP lysis buffer. Lysates were eluted off beads and denatured with 2X NuPAGE LDS Sample Buffer (Life Technologies) containing 5% $\beta$-mercaptoethanol (Sigma-Aldrich) for 15 min at 70°C.

For mass spectrometry, the lysates were run on an SDS–PAGE gel and then stained for 1 h in a 15-cm plate with Novex SimplyBlue SafeStain (Thermo Fisher Scientific) and washed overnight with water at RT. Bands were visualized on Odyssey CLx LI-COR Imaging System and excised with a clean razor blade. Single, excised bands for protein identification were analyzed by LC-MS/MS by MS Bioworks.

For knockdown-rescue biochemical analysis, three to four wells of a six-well plate were pooled together and washed with cold PBS++, lysed in cold with cold 50 mM Tris–HCl, pH 7.4, 150 mM NaCl, 1% Triton X-100, 0.1% SDS, 0.5% sodium deoxycholate, and 1.5X Halt Protease and Phosphatase Inhibitor Cocktail (Thermo Fisher Scientific), scraped, and needle-lysed 12-15X using a 1-ml syringe and 22G needle. Lysates were incubated on ice for 20 min, then centrifuged for 10 min at 4°C at 15,000$g$. After measurement of protein concentration via BCA protein assay (Prometheus), lysates were incubated with ChromoTek GFP-Trap Agarose beads (Proteintech) for 1 h at 4°C to enrich the mEmerald-expressing fusion construct. Beads were spun down and washed 3X with lysis buffer. Lysates were eluted off beads and denatured with 2X NuPAGE LDS Sample Buffer (Life Technologies) containing 5% $\beta$-mercaptoethanol (Sigma-Aldrich) for 5 min at 100°C.

### Image processing and analysis

Relative polarization was assessed by first segmenting cells based on a junctional marker (ZO-1 or VE-cadherin) using a custom MATLAB script, and each cell was split into quadrants such that the length of the cell was split into four equal-length sections. Then, in ImageJ, the integrated density of each quadrant was normalized to the expected integrated density for a region of that size if the signal was evenly distributed across the whole cell. In this way, a relative polarization above 1 indicates a higher concentration of fluorescent signal than expected, and conversely, a value below 1 indicates a lower concentration than expected from an even distribution.

To determine internalization of Notch1 ECD, cells were manually segmented and the integrated density of internalized Notch1 normalized to cell area was measured. To analyze the distribution of Notch1 ECD and ICD immunostaining in downstream domains, a 50-micron line was drawn equally distributed across the cell–cell interface. Fluorescence intensity along this line was quantified via line scan in ImageJ and normalized to the maximal intensity for each line. Within an independent experiment, only regions of interest with similar cell density were compared across conditions. Unless otherwise mentioned, all representative images are intensity-matched across conditions for data visualization.

## Statistical analysis

Sample sizes and *P*-values are reported in each of the corresponding figure legends. Statistical analyses were performed in GraphPad Prism 10. Data distribution was assumed to be normal, but this was not formally tested. Unless otherwise noted, graphs show the mean ± SD. When experiments involved only a single pair of conditions, statistical differences between the two sets of data were analyzed with a two-tailed, unpaired *t* test assuming unequal variances. For datasets containing more than two samples, one-way ANOVA with a classical Bonferroni multiple-comparison post-test was used to determine adjusted *P*-values. Images are representative of at least three independent experiments. Experiments were not randomized, and the investigators were not blinded during data analysis.

# Data Availability

All data are included in the article or supplementary data (Table S2).

# Supplementary Information

# Acknowledgements

We thank Chloe Whitworth and members of the Kutys laboratory for feedback. This work was supported by grants from the NIH (GM150987, AG072232, GM142944), the Leducq Foundation (21CVD03), and NIH Shared Equipment Grant (S10OD028611). T Singh was supported by an NIH Predoctoral Fellowship (HL162520). KA Jacobs was supported by a TRDRP Predoctoral Fellowship (T33DT6442) and an NSF GRFP (2038436).

## Author Contributions

T Singh: data curation, formal analysis, funding acquisition, investigation, methodology, and writing—original draft, review, and editing.
KA Jacobs: data curation, investigation, and methodology.
WJ Polacheck: funding acquisition, methodology, and writing—review and editing.
ML Kutys: conceptualization, resources, data curation, formal analysis, supervision, funding acquisition, validation, investigation, visualization, methodology, project administration, and writing—original draft, review, and editing.

## Conflict of Interest Statement

The authors declare that they have no conflict of interest.

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
