## [Reviewer comments · Life Science Alliance]

The Notch1 intracellular domain orchestrates mechanotransduction of fluid shear stress

Tania Singh, Kyle Jacobs, William Polacheck, and Matthew Kutys

DOI: <https://doi.org/10.26508/lsa.202503599>

Corresponding author(s): Matthew Kutys, University of California, San Francisco

Review Timeline:

Submission Date:	2025-12-15
Editorial Decision:	2025-12-19
Revision Received:	2026-01-16
Editorial Decision:	2026-01-22
Revision Received:	2026-01-22
Accepted:	2026-01-23

Scientific Editor: Tim Fessenden

Transaction Report:

Please note that the manuscript was previously reviewed at another journal and the reports were taken into account in the decision-making process at *Life Science Alliance*.

Reviews

Reviewer #1 Review

Comments to the Authors (Required):

This manuscript extends our understanding of the mechanism of shear stress activation of NOTCH1. It implicates N1ICD interaction with annexin A2 and caveolar proteins.

1. Several data panels require quantitation from multiple experiments and statistical analysis. They include Fig 1E, Fig 2A.
2. Fig. 2A. A single slice is shown. Please show multiple to indicate the range of distributions that occur, and the proportion that show NOTCH1 ECD localised at discrete apical interfaces.
3. Fig 3C. It is uncertain from the data presented whether rhDLL4 influences N1 V1754 in these conditions. N1 V1754 is enhanced by rhDLL4 in two of three experiments. Further experimentation should be carried out to elucidate this.
4. Fig 4A. I did not find a complete list of interacting proteins which should be included. Did Caveolins show the strongest enrichment?
5. In vivo validation would accentuate this study

Minor comments

Introduction - The term laminar flow is not specific enough e.g. Oscillatory flow is usually laminar. Unidirectional flow or unidirectional shear stress is better.

DLL4 protein should be in capitals, and mRNA in capitalised italics

Describe abbreviations of upstream, downstream in figure legends

Reviewer #2 Review

Comments to the Authors (Required):

In this manuscript, Sing et al provide evidence for the mechanism of endothelial NOTCH1 activation by fluid shear stress. The authors first confirm previous results that shear stress stimulates NOTCH redistribution into polarized microdomains and is activated by DLL4. They then test ligands and intracellular interaction partners to determine critical components of shear-induced NOTCH polarization and activation. Results indicate that a NOTCH1 construct lacking the intracellular domain does not re-distribute to polarized domains, arguing for its importance in their assembly. The authors then identify Annexin A2 and CAV1 as flow responsive NOTCH1 interaction partners and assay their roles in NOTCH signaling. Dll4, Annexin A2, and CAV1 are shown to be dispensable for NOTCH1 polarization, but significant mediators of NOTCH1 cleavage.

This study has the capacity to resolve some interesting mechanistic details regarding endothelial NOTCH signaling. However, this manuscript needs to address some concerns.

- 1) Though the authors establish a mild biochemical link between NOTCH to CAV1, results also show that they act independently to organize a polarized membrane domains and facilitate NOTCH cleavage. A critical functional link hinges on Figure 4N, which claims to show that γ -secretase's association with DRMs is modified by shear and dependent upon CAV1. However, my interpretation of the data is that there are no detectable responses in γ -secretase localization. These data need to be quantified and shown to be meaningful.
- 2) NOTCH clustering vs endocytosis. The author's pulse-chase experiments claim to assay endocytosis, but they may just reflect receptor clustering. In fact their extended supplemental video 2 shows that the anti-NOTCH ECD clusters undergo multiple fusion and fission events, which are not characteristic of endocytic vesicles. This may be problematic for the interpretation of Figure 2G-I and Figure 4F-G. With more detailed experiments, such as a biochemical assay, the authors will be able to report more convincing endocytosis results.

Other comments:

- a) Fig 3E-F. I fail to understand why the authors stained these cells with the intracellular domain antibody. The Δ ICD construct is showing significant intracellular localization, which makes it difficult to use emerald fluorescence to determine that there is no polarization. Please repeat with the ECD antibody, if possible.
- b) FigS1 has some mislabeled panels. The panel "S1H" argues that it shows membrane localization of the authors' NOTCH construct, but it appears to show that they fail to reproducibly achieve NOTCH-ECD staining.

- c) Co-immunoprecipitations in Figure 4B need to be repeated with negative control conditions. These are notoriously sticky interaction partners.
- d) It seems odd that the second quadrant has significant NOTCH polarization in Fig 4H. Please ensure that the stats are correct.
- e) Fig2i and 4l show that the NOTCH quantification experiments are not as precise as those shown in Fig 4g. This makes it impossible to determine that CAV1 and DLL4 do not influence NOTCH clustering, as argued by the authors. This discrepancy in the assay should be addressed.
- f) Dermal microvascular endothelial cells are a mixed population of ECs. The authors should specify if these are adult or neonatal cells and what fraction are lymphatic.
- g) The sgRNA sequences listed in materials and methods sometimes include the cloning sequence "CACCG." The authors should remove those nucleotides.
- h) Be sure to provide citations when mentioning "previously described."
- i) The order of figure panels is not well aligned with their description in the text. Fig4i is an example.
- j) One of the Fig3C blots is missing from the source material file.

Reviewer #3 Review

Comments to the Authors (Required):

In this manuscript, the authors study the mechanism of Notch activation in flow. As previously by Mack et al. and Giese et al., they find that Notch1 polarizes downstream when exposed to shear stress. They find that DLL4 ablation is not required for polarization, but the ICD Is required for polarization. They show a lack of trans-endocytosis of the ECD. They then show that Notch associates with Annexin2 and Cav1 in the presence of flow, but these are not required for mechanotransduction of Notch.

Major

For Figure 1C, the Jag1 KO static/flow are often done on a different blot than their scrambled controls. They need to be run on the same blot. You cannot compare signal from one blot to another. Please rerun these blots as one gel.

In figure 1C/D, there is still a doubling of the N1 1754 staining by Western blot in the presence of shear stress. The results are not statistically significant but there is only an n=3. Please increase these experiments at least to an n=5. They are the basis of the whole paper, so they need to be on sound footing.

There are two references for the CRISPR technique. One is for an immortalised epithelial cell line. The second reference is for a biorxiv deposited thesis (for which the paper has yet to be published). The text refers to a "Stable CRISPR-modified" of primary endothelial cells. My understanding is that CRISPR is quite difficult in primary cell lines. What do the authors mean by "stable"? At what passage were the cells modified? At what passage number were they used in flow experiments? The text says passage 2 to 8 for cell culture, but specifically for the CRISPR cells, which passage number was it?

Also for figure 1D, the same data is presented twice, but given that the same values are used for control, this should be one graph. They are not separate independent experiment.

A one-way ANOVA is not appropriate for figure 1B, 1D, 4K. These should be analysed as two-way ANOVA's.

A time series, such as those presented in 1H, 1K, 1L, 2C, 3F, 4H, 4M need to be analysed by a mixed effect model. It is not clear what the "stars" indicate. What should be indicated is whether one curve is statistically different than the other curve (not whether individual points are statistically different than values at time t=0)

It is not clear why the LEC results are included. Because they are cell type that apparently do not cluster DS in response to flow and do not activate Notch in response to flow? It is not clear that the level of shear stress is even appropriate for a LEC. Choi et al. found that 24h of laminar shear stress (2 dyn/cm²) reduced the levels of N1 ICD. I believe the current study uses 20 dyn/cm². Why would a LEC be responsive at a level of shear stress that it is never exposed to? I personally would remove the LECs from the story entirely. Alternatively, could the authors expose LECs to 2 dyn/cm² and see if they activate Notch or cluster Notch1.

The results in figure 2D are interesting but confusing. Is the ligand being internalized with the ECD or does the ligand dissociate before internalisation? Does inhibition of the S2 cleavage change the internalisation observed in figure 2D. This would be preferable with more specific inhibitors of ADAM such as GW280264X and GI254023X, then BB94. It is not clear how, if the S2 cleavage occurs, it would be possible for the ECD to enter the receiving cell. This cleavage step is needed for conformational changes to expose the S3 site. Possibly

under flow confirmational changes can occur in a different manner?

The authors state that supplemental video shows internalisation. They also state the results show cis-endocytosis. While I agree that the results negate trans-endocytosis, they do not show internalisation per se. The movies cannot differentiate between extra cellular and intracellular. The authors need to use a proper internalization assay (see comments below on pulse chase).

On a similar note, it is not clear to me where the "chase" is in the pulse-chase. Normally, you would add a labelled antibody and then use unlabelled antibody as the "chase" to remove all non-internalised antibody. The methods say the antibody was added at the beginning of the experiment, but they were never competed off the cell. I don't think a wash with PBS will remove bound antibody. This is just live cell labelling of ECD that was extracellular at the beginning of the experiment. But there is no proof that it is intracellular at the end of the experiment. The authors need to stain for the "pulse" both with and without cell permeabilization. Or do a treatment before permeabilizing the cells to remove all surface proteins, and then stain for their labelling antibody. Alternatively, a method of fractionation showing the pulsed antibody is no longer membrane bound is also possible.

Does ANXA2-ko affect total Notch1 levels? Does CAV1-KO affect total Notch1 levels?

In figure 3D, the authors use mDLL4 expressing cells and Notch1- Δ ICD expressing cells at different proportions. Are the mDLL4 cells needed? Is this not just a dilution effect? The differing amount of N1 V1754 is due to more cells having ligand independent activation. The author should go culture with a non-ligand expressing cell type (even if it is non-endothelial) to see if they get a different result. If they don't, then it is just a dilution effect.

Figure 4B - please quantify the CoIP. Please investigate whether the Notch1- Δ ICD associates with Annexin2 under control and flow conditions.

Could the authors show the total levels of Notch1 with Annexin2 KO.

In figure 1H, the Y axis is very different than the other alignment figures (even for the scrambled control)? Is this curve statistically different than a straight line, for control of KO? All the others polarization curves that are shown have a range of 0.8 to 1.2 from one end to the other. In figure 1H, it goes from 0.95 to 1.05. I think that experiment just failed to polarise, even in the control conditions. Could you repeat it? I don't think either scrambled or ANXA2-KO is statistically different than a straight line (i.e. no polarization).

Please quantify (and normalize) the amount of protein observed in the detergent resistant fragments in Figure 4N. Please quantify total Notch 1 as well with Cav1 KD.

The authors refer to results showing that Cav1 and Notch1 are organised into microdomains to which the secretases are present. This is based on one of the author's 2022 papers for ligand-based Notch signalling. Are the secretases under flow in the detergent resistant fraction? The authors have not shown the same mechanism is occurring here.

The paper cited, Kwak 2022, shows that the full length receptor is excluded from these microdomains due to the large size of the ECD and that the cleavage of the ECD allows it to enter the microdomain. The author's current results indicate that the extracellular domain is not cleaved however? Or doesn't seem to be cleaved because it is internalised? The authors need to establish whether the ECD is cleaved or not (and whether it is actually internalised).

In the discussion, the authors imply that the cleavage events occurring in endosomes could explain their results (and I agree, they could explain their results). The authors should isolate endosomes or co-stain for Rab proteins to show internalisation of the ligand/receptor complex in the presence of flow. This could also be used as "proof" of internalization.

Minor

For figure 1, how long was shear stress applied? Supplemental figure 1 has both 1h and 24h, but what was done for figure 1 is not specified.

US and DS should be defined in the figure legend.

In figure 2A, it is not clear to me what is a labelled in white. Is that they ICD? In the zoomed-out image (first

panel of 2A), why are white and cyan not shown? If it is just to show that the purple cell is upstream of the green cell, why also show this in the cartoon?

SNAP tagged TMD is S1G, not S1F as indicated in the text.

Why were investigators not blinded during analysis?

It is not clear to me why the N1p-Cre or the synthetic notch receptors are mentioned in the introduction to the results of figure 3. The results use neither strategy.

December 19, 2025

Re: Life Science Alliance manuscript #LSA-2025-03599-T

Dr. Matthew L. Kutys
University of California, San Francisco
Department of Cell and Tissue Biology
513 Parnassus Avenue, HSW-613
San Francisco, CA 94143

Dear Dr. Kutys,

Thank you for submitting your manuscript entitled "The Notch1 intracellular domain orchestrates mechanotransduction of fluid shear stress" to Life Science Alliance. The manuscript and the accompanying reviews from another journal have now been evaluated. This work introduces a novel means by which shear flow activates Notch signaling in vascular endothelial cells separately from juxtacrine cell-cell activation and trans-endocytosis. As noted by Reviewers 1 and 2, this work makes inroads into how endothelial cells may use Notch signaling in their adaptation to shear flow. While all reviewers suggested new data to strengthen and extend the molecular mechanism by which this signaling axis operates, a major extension of this work is not required for further consideration at LSA. We invite you to submit a revised manuscript addressing the following Reviewer comments, which we will evaluate internally without further reviewer input.

- Either repeat the assay in Figure 4N and quantify the results or remove the claim that presenilin-1 is enriched in detergent-resistant lipid fractions with Notch, per concerns by Reviewers 2 and 3.
- Discuss the possibility of Notch clustering in addition to endocytosis as raised by Reviewers 2 and 3.
- Either report the total levels of Notch1 in Anx2a and Cav1 KO cells or state that the levels were not evaluated, per Reviewer 3.
- Address Reviewer 3 concerns on Figure 1C/D: show the separate control blot for Fig 1C if any, or place quantifications on the same graph if these were run together.
- Discuss the role of S2 cleavage shown in Figure 2D and how this event fits with your overall model, per Reviewer 3.
- Verify that appropriate statistical tests are used throughout and improve these if needed, per Reviewer 3.
- Correct all text and figure presentation errors noted by reviewers.
- Provide the full mass spectrometry data (for instance an accession number to an online repository), per Reviewer 1 and to align with LSA publication requirements.

I would be happy to discuss the revision in more detail via email or phone/videoconferencing. Please let me know which option you prefer, if any.

While you are revising your manuscript, please also attend to the below editorial points to help expedite the publication of your manuscript. Please direct any editorial questions to the journal office. When submitting the revision, please include a letter addressing the reviewers' comments point by point.

Thank you for this interesting contribution to Life Science Alliance. We are looking forward to receiving your revised manuscript.

Sincerely,

- A letter addressing the reviewers' comments point by point.
- An editable version of the final text (.DOC or .DOCX) is needed for copyediting (no PDFs).

B. MANUSCRIPT ORGANIZATION AND FORMATTING:

Reviewer #1 (Comments to the Authors (Required)):

This manuscript extends our understanding of the mechanism of shear stress activation of NOTCH1. It implicates N1ICD interaction with annexin A2 and caveolar proteins.

1. Several data panels require quantitation from multiple experiments and statistical analysis.

They include Fig 1E, Fig 2A.

Figure 1E is quantified in Figure 1F. Downstream polarization of Notch1 ECD is quantified as a metric of Notch1 polarization throughout the manuscript, including Figs. 1K,L; 2B; 4H,M.

2. Fig. 2A. A single slice is shown. Please show multiple to indicate the range of distributions that occur, and the proportion that show NOTCH1 ECD localised at discrete apical interfaces.

The orthogonal projection in Fig. 2A is not a single slice and is representative of the apical-basal range of distribution as requested.

3. Fig 3C. It is uncertain from the data presented whether rhDLL4 influences N1 V1754 in these conditions. N1 V1754 is enhanced by rhDLL4 in two of three experiments. Further experimentation should be carried out to elucidate this.

We have modified our description of this claim in the results. "In contrast, activation of Notch1-ΔICD did not increase in response to flow and was diminished when plated on immobilized rhDll4 substrates (Fig. 3C)". Coupled with the observation that Cav1 deletion diminishes N1 V1754 on rhDLL4, we believe this claim is sound and important to include.

4. Fig 4A. I did not find a complete list of interacting proteins which should be included. Did Caveolins show the strongest enrichment?

We have included a supplementary excel table which includes the full mass spectrometry dataset for the band excision microscopy.

5. In vivo validation would accentuate this study

In vivo experiments are outside the scope of this study and would not materially change the conclusions.

Minor comments

Introduction - The term laminar flow is not specific enough e.g. Oscillatory flow is usually laminar. Unidirectional flow or unidirectional shear stress is better.

As requested, we have fixed all instances in the text.

DLL4 protein should be in capitals, and mRNA in capitalised italics
As requested, we have fixed all instances in the text.

Describe abbreviations of upstream, downstream in figure legends
As requested, this has been added to all legends.

Reviewer #2 (Comments to the Authors (Required)):

In this manuscript, Sing et al provide evidence for the mechanism of endothelial NOTCH1 activation by fluid shear stress. The authors first confirm previous results that shear stress stimulates NOTCH redistribution into polarized microdomains and is activated by DLL4. They then test ligands and intracellular interaction partners to determine critical components of shear-induced NOTCH polarization and activation. Results indicate that a NOTCH1 construct lacking the intracellular domain does not re-distribute to polarized domains, arguing for its importance in their assembly. The authors then identify Annexin A2 and CAV1 as flow responsive NOTCH1 interaction partners and assay their roles in NOTCH signaling. Dll4, Annexin A2, and CAV1 are shown to be dispensable for NOTCH1 polarization, but significant mediators of NOTCH1 cleavage.

This study has the capacity to resolve some interesting mechanistic details regarding endothelial NOTCH signaling. However, this manuscript needs to address some concerns.

1) Though the authors establish a mild biochemical link between NOTCH to CAV1, results also show that they act independently to organize a polarized membrane domains and facilitate NOTCH cleavage. A critical functional link hinges on Figure 4N, which claims to show that γ -secretase's association with DRMs is modified by shear and dependent upon CAV1. However, my interpretation of the data is that there are no detectable responses in γ -secretase localization. These data need to be quantified and shown to be meaningful.

We have removed the claim that γ -secretase is enriched in detergent-resistant lipid fractions with Notch1 by flow and discuss this possible mechanism and its relation to caveloin-1 in the Discussion. (Discussion paragraph #2).

2) NOTCH clustering vs endocytosis. The author's pulse-chase experiments claim to assay endocytosis, but they may just reflect receptor clustering. In fact their extended supplemental video 2 shows that the anti-NOTCH ECD clusters undergo multiple fusion and fission events, which are not characteristic of endocytic vesicles. This may be problematic for the interpretation of Figure 2G-I and Figure 4F-G. With more detailed experiments, such as a biochemical assay, the authors will be able to report more convincing endocytosis results.

While endocytic vesicles are indeed able to fuse, we acknowledge that receptor clustering was not investigated and now discuss clustering as a possible step in shear-mediated Notch1 mechanotransduction. (Discussion paragraph #2).

Other comments:

a) Fig 3E-F. I fail to understand why the authors stained these cells with the intracellular domain antibody. The Δ ICD construct is showing significant intracellular localization, which makes it difficult to use emerald fluorescence to determine that there is no polarization. Please repeat with the ECD antibody, if possible.

Expression of Notch1-WT displays significant polarization as quantified by the mEmerald signal fused to the expressed receptor. This is not the case for the Δ ICD construct, underlying the importance of the domain. The ICD antibody was used as a control to demonstrate the presence or absence of the ICD in each case. It is no surprise that the immunostaining shows polarization in the Notch1-WT cells, but not in the Δ ICD construct.

b) FigS1 has some mislabeled panels. The panel "S1H" argues that it shows membrane localization of the authors' NOTCH construct, but it appears to show that they fail to reproducibly achieve NOTCH-ECD staining.

The figure is appropriately labeled. The staining of Notch1-WT is differs slightly due to the rapid internalization of the expressed construct.

c) Co-immunoprecipitations in Figure 4B need to be repeated with negative control conditions. These are notoriously sticky interaction partners.

We identified these proteins by band excision mass spectrometry and confirmed via co-immunoprecipitation western blot. Additionally, the abundance of these co-immunoprecipitations, as well as immunofluorescence co-localization are reproducibly influenced by shear stress.

d) It seems odd that the second quadrant has significant NOTCH polarization in Fig 4H. Please ensure that the stats are correct.

Here, significance is compared to an unpolarized line for each condition, not the line representing the other condition. We have clarified this in the legends and methods.

e) Fig2i and 4l show that the NOTCH quantification experiments are not as precise as those shown in Fig 4g. This makes it impossible to determine that CAV1 and DLL4 do not influence NOTCH clustering, as argued by the authors. This discrepancy in the assay should be addressed.

We provide detailed methodology for our internalization assay, which was applied uniformly to reveal a significant difference for annexin A2 knockout cells, but not Cav1 or DLL4 knockout cells.

f) Dermal microvascular endothelial cells are a mixed population of ECs. The authors should specify if these are adult or neonatal cells and what fraction are lymphatic.

This is correct. However, the cells used in this study were sorted cell populations from Lonza Inc with a description of Prox1+ cells for each population, ensuring their purity for BEC and LEC subtypes. We have added these details to the methods.

g) The sgRNA sequences listed in materials and methods sometimes include the cloning sequence "CACCG." The authors should remove those nucleotides.

As requested, we have fixed all instances in the text.

h) Be sure to provide citations when mentioning "previously described."

This instance has been removed from the methods as part of the detergent membrane extraction details.

i) The order of figure panels is not well aligned with their description in the text. Fig4i is an example.

Fig. 4I begins the section on caveolin-1 one and is properly placed.

j) One of the Fig3C blots is missing from the source material file.

We have corrected the blot in the source material file.

Reviewer #3 (Comments to the Authors (Required)):

In this manuscript, the authors study the mechanism of Notch activation in flow. As previously by Mack et al. and Giese et al., they find that Notch1 polarizes downstream when exposed to shear stress. They find that DLL4 ablation is not required for polarization, but the ICD is required for polarization. They show a lack of trans-endocytosis of the ECD. They then show that Notch associates with Annexin2 and Cav1 in the presence of flow, but these are not required for mechanotransduction of Notch.

Major

For Figure 1C, the Jag1 KO static/flow are often done on a different blot than their scrambled controls. They need to be run on the same blot. You cannot compare signal from one blot to another. Please rerun these blots as one gel.

All static and flow experiments quantifications were made from the same gel and we have updated our quantification to a single graph to reflect this.

In figure 1C/D, there is still a doubling of the N1 1754 staining by Western blot in the presence of shear stress. The results are not statistically significant but there is only an n=3. Please increase these experiments at least to an n=5. They are the basis of the whole paper, so they need to be on sound footing.

It is unclear what the reviewer is referring to. Scramble and JAG1 KO cells both show statistical increases in cleaved Notch1. There is no statistical increase in Notch1 cleavage in DLL4 KO.

There are two references for the CRISPR technique. One is for an immortalised epithelial cell line. The second reference is for a biorxiv deposited thesis (for which the paper has yet to be published). The text refers to a "Stable CRISPR-modified" of primary endothelial cells. My understanding is that CRISPR is quite difficult in primary cell lines. What do the authors mean by "stable"? At what passage were the cells modified? At what passage number were they used in flow experiments? The text says passage 2 to 8 for cell culture, but specifically for the CRISPR cells, which passage number was it?

The reviewer has omitted an additional reference to (Polacheck and Kutys et al. 2017) which is the original paper in which we developed and validated the CRISPR method in primary human endothelia to discover the connection between Notch1 and flow. We have added additional data to the methods describing this technique.

Also for figure 1D, the same data is presented twice, but given that the same values are used for control, this should be one graph. They are not separate independent experiment.

The reviewer is correct, and we have corrected the graph.

A one-way ANOVA is not appropriate for figure 1B, 1D, 4K. These should be analysed as two-way ANOVA's.

In these instances, we are reporting single variant significance (within a given genetic condition) and therefore a one-way ANOVA is appropriate. We have assured statistical analyses are appropriate throughout the manuscript.

A time series, such as those presented in 1H, 1K, 1L, 2C, 3F, 4H, 4M need to be analysed by a mixed effect model. It is not clear what the "stars" indicate. What should be indicated is whether one curve is statistically different than the other curve (not whether individual points are statistically different than values at time $t=0$)

This appears to be a misunderstanding by the reviewer. These data are not time series, but rather spatial maps of Notch1 distribution along the length of the cell. The appropriate statistical tests have been used in comparing the statistical deviation of each quadrant to an unpolarized line distribution. This has been clarified in the legends and methods.

It is not clear why the LEC results are included. Because they are cell type that apparently do not cluster DS in response to flow and do not activate Notch in response to flow? It is not clear that the level of shear stress is even appropriate for a LEC. Choi et al. found that 24h of laminar shear stress (2 dyn/cm²) reduced the levels of N1 ICD. I believe the current study uses 20 dyn/cm². Why would a LEC be responsive at a level of shear stress that it is never exposed to? I personally would remove the LECs from the story entirely. Alternatively, could the authors expose LECs to 2 dyn/cm² and see if they activate Notch or cluster Notch1.

As we explain in the results section, we are not examining the response of lymphatic endothelia to physiologic levels of shear stress, but rather whether Notch1-DLL4 itself is sufficient to operate as a mechanosensor of fluid shear stress. Our conclusion is that because hdLEC and hdBEC express comparable levels of Notch1 and DLL4, yet respond differently to an identical mechanical shear stress challenge, that a unique mechanotransduction mechanism is operating in hdBECs under this condition. We further demonstrate the details of this mechanism being specific to hdBEC in Figure 4.

The results in figure 2D are interesting but confusing. Is the ligand being internalized with the ECD or does the ligand dissociate before internalisation? Does inhibition of the S2 cleavage change the internalisation observed in figure 2D. This would be preferable with more specific inhibitors of ADAM such as GW280264X and GI254023X, then BB94. It is not clear how, if the S2 cleavage occurs, it would be possible for the ECD to enter the receiving cell. This cleavage step is needed for conformational changes to expose the S3 site. Possibly under flow conformational changes can occur in a different manner?

'This cleavage step is needed for conformational changes to expose the S3 site', is not true in all cases as we demonstrate in (Kwak et al., 2022). The removal of the ECD is often needed to navigate steric barriers to access gamma secretase, but its removal is not required if such barriers are bypassed. S2 cleavage is not associated with Notch1 polarization by flow, but we agree its contribution to this mechanism is still unknown and we explore possibilities in the Discussion section.

The authors state that supplemental video shows internalisation. They also state the results show cis-endocytosis. While I agree that the results negate trans-endocytosis, they do not show internalisation per se. The movies cannot differentiate between extra cellular and intracellular. The authors need to use a proper internalization assay (see comments below on pulse chase).

While the Results highlight that the behavior of labeled Notch receptors in the video display many hallmarks of endocytosed vesicles, we acknowledge that extracellular receptor clustering was not investigated and now discuss clustering as a possible step in shear stress-mediated mechanotransduction. (Discussion paragraph #2).

On a similar note, it is not clear to me where the "chase" is in the pulse-chase. Normally, you would add a labelled antibody and then use unlabelled antibody as the "chase" to remove all non-internalised antibody. The methods say the antibody was added at the beginning of the experiment, but they were never competed off the cell. I don't think a wash with PBS will remove bound antibody. This is just live cell labelling of ECD that was extracellular at the beginning of the experiment. But there is no proof that it is intracellular at the end of the experiment. The authors need to stain for the "pulse" both with and without cell permeabilization. Or do a treatment before permeabilizing the cells to remove

all surface proteins, and then stain for their labelling antibody. Alternatively, a method of fractionation showing the pulsed antibody is no longer membrane bound is also possible.

In addition to the discussion noted above, we have altered description of the assay from antibody pulse-chase labeling to antibody pulse labeling.

Does ANXA2-ko affect total Notch1 levels? Does CAV1-KO affect total Notch1 levels?

They do not. Total Notch1 western blots are now provided in Figure 4 for both knockouts.

In figure 3D, the authors use mDLL4 expressing cells and Notch1- Δ ICD expressing cells at different proportions. Are the mDLL4 cells needed? Is this not just a dilution effect? The differing amount of N1 V1754 is due to more cells having ligand independent activation. The author should go culture with a non-ligand expressing cell type (even if it is non-endothelial) to see if they get a different result. If they don't, then it is just a dilution effect.

The molecular weight in the figure indicates that the N1 1754 is reading out activation of only the expressed Notch1- Δ ICD, not total Notch1. Despite decreasing proportions of Notch1- Δ ICD, the proteolysis of N1 1754 increases, suggesting it is not a dilution effect. It is not clear to us what the reviewer is referring to as ligand-independent activation.

Figure 4B - please quantify the CoIP. Please investigate whether the Notch1- Δ ICD associates with Annexin2 under control and flow conditions.

We agree that identifying whether the ICD, alone or in concert with other domains, controls the identified flow-stimulated interactions. Those efforts are planned for follow on studies.

In figure 1H, the Y axis is very different than the other alignment figures (even for the scrambled control)? Is this curve statistically different than a straight line, for control of KO? All the others polarization curves that are shown have a range of 0.8 to 1.2 from one end to the other. In figure 1H, it goes from 0.95 to 1.05. I think that experiment just failed to polarise, even in the control conditions. Could you repeat it? I don't think either scrambled or ANXA2-KO is statistically different than a straight line (i.e. no polarization).

All experimental sets were run in parallel to a control condition. There may be inherent assay variability, but both lines are statistically significant from an unpolarized line. Additionally, N1 ECD antibody pulse labeling of annexin A2 knockout cells displays polarization, further suggesting there is no defect with receptor polarization in these cells.

Please quantify (and normalize) the amount of protein observed in the detergent resistant fragments in Figure 4N. Please quantify total Notch 1 as well with Cav1 KD.

The authors refer to results showing that Cav1 and Notch1 are organised into microdomains to which the secretases are present. This is based on one of the author's 2022 papers for ligand-based Notch signalling. Are the secretases under flow in the

detergent resistant fraction? The authors have not shown the same mechanism is occurring here. The paper cited, Kwak 2022, shows that the full length receptor is excluded from these microdomains due to the large size of the ECD and that the cleavage of the ECD allows it to enter the microdomain. The author's current results indicate that the extracellular domain is not cleaved however? Or doesn't seem to be cleaved because it is internalised? The authors need to establish whether the ECD is cleaved or not (and whether it is actually internalised). In the discussion, the authors imply that the cleavage events occurring in endosomes could explain their results (and I agree, they could explain their results). The authors should isolate endosomes or co-stain for Rab proteins to show internalisation of the ligand/receptor complex in the presence of flow. This could also be used as "proof" of internalization.

We have removed the claim that presenilin-1 is enriched in detergent-resistant lipid fractions with Notch1 and introduce this possibility as a mechanism in the discussion. Additionally, we discuss the possibilities for the role and site of ECD cleavage (Discussion paragraph #2).

Minor

For figure 1, how long was shear stress applied? Supplemental figure 1 has both 1h and 24h, but what was done for figure 1 is not specified.

As requested, this detail has been added.

US and DS should be defined in the figure legend.

As requested, This has been corrected in all legends.

In figure 2A, it is not clear to me what is a labelled in white. Is that they ICD? In the zoomed-out image (first panel of 2A), why are white and cyan not shown? If it is just to show that the purple cell is upstream of the green cell, why also show this in the cartoon?

The cartoon has been modified further for figure clarity.

SNAP tagged TMD is S1G, not S1F as indicated in the text.

This has been corrected.

Why were investigators not blinded during analysis?

Blinding was not a part of the original data collection and analysis.

It is not clear to me why the N1p-Cre or the synthetic notch receptors are mentioned in the introduction to the results of figure 3. The results use neither strategy.

N1p-Cre or the synthetic notch receptors approach provides the rationale and strategy for replacing the ICD at Arg1764, as described in the text. We have removed synNotch mentions for clarity.

January 22, 2026

RE: Life Science Alliance Manuscript #LSA-2025-03599-TR

Dr. Matthew L. Kutys
University of California, San Francisco
Department of Cell and Tissue Biology
513 Parnassus Avenue, HSW-613
San Francisco, CA 94143

Dear Dr. Kutys,

Thank you for submitting your revised manuscript entitled "The Notch1 intracellular domain orchestrates mechanotransduction of fluid shear stress". We have evaluated your manuscript in light of our previous requests to address key reviewer concerns. As these have all been resolved, we would be happy to publish your paper in Life Science Alliance pending final revisions necessary to meet our formatting guidelines.

MANUSCRIPT ORGANIZATION AND FORMATTING:

To avoid unnecessary delays in the acceptance and publication of your paper, please read the following information carefully. Full guidelines are available on our Instructions for Authors page, <https://www.life-science-alliance.org/authors>

- Please remove the separate supporting information file.
- Please add your main, supplementary figure, and table legends to the main manuscript text after the references section.
- Please add a Category for your manuscript in our system.
- Please add the X and Bluesky handles of your host institute/organization, as well as your own, and/or one of the authors, in our system.
- Please consult our manuscript preparation guidelines <https://www.life-science-alliance.org/manuscript-prep> and make sure your manuscript sections are in the correct order.
- Please add the following sections to your manuscript text: Conflict of Interest, Author Contributions, Data Availability (again see <https://www.life-science-alliance.org/manuscript-prep>).
- There is a call-out for figure 4P instead of 4O. Please correct to match the legend and the actual figure.
- Please add callouts for Figure S1F, G, and I.

LSA encourages authors to provide a 30-60 second video where the study is briefly explained. We will use these videos on social media to promote the published paper and the presenting author (for examples, see <https://docs.google.com/document/d/1-UWCfbE4pGcDdcgzcmiuJI2XMBJnxKYeqRvLLrLSo8s/edit?usp=sharing>). Corresponding or first-authors are welcome to submit the video. Please submit only one video per manuscript. The video can be emailed to contact@life-science-alliance.org

FINAL FILES:

The following items are required for acceptance.

The license to publish form must be signed before your manuscript can be sent to production. A link to the license to publish form will be available to the corresponding author only. Please take a moment to check your funder requirements.

Thank you for your attention to these final processing requirements. Please revise and format the manuscript and upload materials as soon as you are able.

Thank you for this interesting contribution to the literature. We look forward to publishing your paper in Life Science Alliance.

Sincerely,

January 23, 2026

RE: Life Science Alliance Manuscript #LSA-2025-03599-TRR

Dr. Matthew L. Kutys
University of California, San Francisco
Department of Cell and Tissue Biology
513 Parnassus Avenue, HSW-613
San Francisco, CA 94143

Dear Dr. Kutys,

Thank you for submitting your revised Research Article entitled "The Notch1 intracellular domain orchestrates mechanotransduction of fluid shear stress". It is a pleasure to let you know that your manuscript is now accepted for publication in Life Science Alliance. Congratulations on this interesting work!

Reviews, decision letters, and point-by-point responses associated with peer-review at Life Science Alliance will be published, alongside the manuscript. If you want to opt out of having your point-by-point responses displayed, please let us know immediately.

DISTRIBUTION OF MATERIALS:

Again, congratulations on a very nice paper. I hope you found the review process to be constructive and are pleased with how the manuscript was handled editorially. We look forward to future exciting submissions from your lab.

Sincerely,
